# Feline Soft Tissue Sarcomas: A Review of the Classification and Histological Grading, with Comparison to Human and Canine

**DOI:** 10.3390/ani12202736

**Published:** 2022-10-12

**Authors:** Melanie Dobromylskyj

**Affiliations:** Histopathology Department, Finn Pathologists, One Eyed Lane, Weybread, Diss IP21 5TT, Norfolk, UK; melanie.dobromylskyj@finnpathologists.com

**Keywords:** feline, cat, sarcoma, fibrosarcoma, tumour, soft tissue, injection site, grading, prognosis

## Abstract

**Simple Summary:**

Soft tissue sarcomas are a common form of cancer arising in the skin and connective tissues of domestic cats. Soft tissue sarcomas encompass a group of different histological subtypes of tumours, which can behave in a range of different ways in the patient. In dogs and in humans, this group of tumours can be given a histological score (“grade”) at the time of diagnosis, which is prognostic, but there is no equivalent, well-established grading system for these tumours in cats. This review looks at soft tissue sarcomas in terms of which histological subtypes of tumour should be included in this group, and how pathologists approach their grading, comparing feline tumours with their human and canine counterparts.

**Abstract:**

Soft tissue sarcomas are one of the most commonly diagnosed tumours arising in the skin and subcutis of our domestic cats, and are malignant neoplasms with a range of histological presentations and potential biological behaviours. However, unlike their canine and human counterparts, there is no well-established histological grading system for pathologists to apply to these tumours, in order to provide a more accurate and refined prognosis. The situation is further complicated by the presence of feline injection site sarcomas as an entity, as well as confusion over terminology for this group of tumours and which histological types should be included. There is also an absence of large scale studies. This review looks at these tumours in domestic cats, their classification and histological grading, with comparisons to the human and canine grading system.

## 1. Introduction 

Soft tissue sarcomas (STS) are a group of commonly diagnosed tumours in domestic cats. These are malignant neoplasms, with a range of histological subtypes and also of potential biological behaviours [1,2,3]. In this review, the sometimes confusing terminology surrounding these tumours is discussed, together with the prevalence of these tumours in cats and their clinical behaviour in dogs and cats. The aims of this review are also to explore the different histological subtypes included in the group “soft tissue sarcomas” in cats, dogs and in humans, the grading scheme applied to canine and human STSs and the grading system which has been proposed for feline STS. The review also looks at feline injection site sarcomas, the role of immunohistochemical markers, and the subjectivity of histological grading.

## 2. Terminology

The terminology surrounding these tumours can be confusing, partly because there are a myriad of terms used in the published literature for these neoplasms. Mesenchymal neoplasms are those which arise from mesenchymal cell types, and these can be either malignant or benign; the term sarcoma is typically used to denote malignant neoplasms of mesenchymal origin. Not all mesenchymal tumours arise from soft tissues though; they may, for example, arise from other tissues such as cartilage or bone. To slightly confuse the issue, mesenchymal tumours arising from soft tissues may demonstrate cartilage and/or bone formation (for example, extra-skeletal chondrosarcoma, or extra-skeletal osteosarcoma [4]. Whilst soft tissue sarcomas may potentially arise at any anatomical location where soft mesenchymal tissue is present, those arising from the cutis and subcutis are the focus of this review.

For a subset of these tumours derived from soft tissues, the traditional name used is soft tissue sarcoma (STS) [5]. However, it has been suggested that “soft tissue tumours” or “tumours of soft tissues” might be more appropriate terms, given that the word “sarcoma” implies malignancy, but that it can sometimes be very difficult to differentiate low grade malignant mesenchymal neoplasms from their benign counterparts [3,6]. Furthermore, for the low grade malignant tumours in particular, these tend not to be life-threatening and to not metastasize [1,2,5,7]. There has not been universal acceptance of this suggested change in terminology to date however, and the name “soft tissue sarcoma” appears to be the most commonly used and widely understood still. 

The most recent publication [6] from the Davis Thompson DVM foundation (a group created to further the education in veterinary and comparative pathology [8]) uses the terminology “soft tissue tumours”; soft tissues for the purposes of that publication are defined as “extra-skeletal connective tissues of the dermis, subcutis and fascia, striated and smooth muscle, vessels, serosal and synovial linings and nerve sheaths.” Within this broad tumour group, some histological types are considered together as a subgroup for the purposes of grading and prognostication, despite having different mesenchymal cell origins; these tumours are generally considered together due to their similar histological features and biological behaviours in the patient. Diagnosis is typically made by histological assessment and differentiating between the various histological subtypes with confidence is not necessarily always possible by light microscopy alone [2,9,10,11]. A panel of immunohistological stains are oftentimes needed to diagnose a more specific subtype but is generally not warranted in a routine diagnostic setting given the similar biological behaviours seen and the cost implications [1,9,10,11,12,13]. 

## 3. Prevalence of Soft Tissue Sarcomas in Cats

A retrospective study on the longevity and mortality of cats attending primary care veterinary practices in England found that neoplasia was the fourth most common cause of death in cats [14]. Of these, tumours arising in the skin and subcutis are the most common form of neoplasia diagnosed in cats [15,16] and of those, more than half are malignant [17,18]. 

It has long been known that STSs, encompassing fibrosarcomas and nerve sheath tumours (NST) amongst other histological subtypes, are one of the most commonly diagnosed categories of neoplasia arising in the skin and subcutis of cats. Large scale studies in the United Kingdom [17], Europe [16,18,19] and the USA [15] have all demonstrated, across the decades and various continents, that fibrosarcomas/STSs consistently rank in the top four when it comes to the most common tumours diagnosed in feline skin and soft tissues. 

The typical age of a cat at the time of presentation for STS is 10–11 years [15,17,19], or “middle aged or older” [16]. However, even in cats under the age of 12 months, malignant mesenchymal neoplasms (i.e., those arising from any mesenchymal cell type, not only soft tissue) have been reported to account for 16% of all tumours in this age group. Those tumours arising from soft tissues specifically accounted for 15% overall, and the skin and soft tissues was the most site at 41% [20]. 

Whilst discussing STS arising in young cats, it is worth mentioning feline sarcoma virus (FeSV), a virus which arises from the combination of feline leukaemia virus (FeLV) pro-viral particles with parts of the infected cat’s own genome. FeSV oncogenes promote the transformation of fibroblasts and result in fibrosarcoma development. These account for only a low percentage of fibrosarcomas in cats, but the tumours do tend to be multicentric, carry increased risk of metastasis and occur in young cats predominantly [21,22].

Ho et al. [17] noted that several breeds, including Persian, Siamese, Burmese and British Blue, had significantly lower odds of developing fibrosarcoma when compared with a non-pedigree cat population.

## 4. Clinical Behaviour of Soft Tissue Sarcomas in Cats and Dogs

STSs as a whole, regardless of species, tend to be locally infiltrative and may often extend along fascial planes; typically they are described as having a ‘pseudocapsule’ and a relatively slow growth rate. In dogs, they tend to have a low to moderate incidence of metastasis [1,2,5,7]. However, their clinical behaviour can vary and post-surgical recurrence was shown to be a relatively common occurrence, for example in cats with fibrosarcomas [23], thought to be due to often having poorly defined margins. STSs can be superficial and mobile, or may infiltrate deeper tissues; they may persist for weeks to months with only minor changes before undergoing rapid growth [24,25,26]. 

The existence of feline injection site sarcomas (FISS) further complicates the picture for cats; these are tumours that arise at previous sites of vaccination (historically known as vaccine-associated fibrosarcoma) or other forms of localised trauma which induce chronic inflammation; a full review of FISS is outside of the scope of this current review, but there are several reviews published [27,28,29] and the European Advisory Board on Cat Diseases provides comprehensive and regularly updated information [30] on FISS. 

## 5. Which Histological Subtypes Should Be Included in the STS Group?

*(a)* 
*human:*


In humans, STSs are considered to be relatively rare, accounting for approximately 1% of all malignant tumours in the adult population [31]. There are over 100 different histological subtypes and the group includes tumours arising most commonly on the trunk, extremities and also in the retroperitoneal space. The most commonly diagnosed include liposarcoma, leiomyosarcoma and undifferentiated pleomorphic sarcoma (UPS); as per the 2020 World Health Organisation classification of soft tissue tumours for humans, these tumours are categorised as adipocytic, fibroblastic or myofibroblastic, fibrohistiocytic, smooth muscle, pericytic, skeletal muscle, vascular, gastrointestinal, nerve sheath origin, chondro-osseous, undifferentiated/unclassified or uncertain [31]. The general preference is still to grade these tumours according to the system originally proposed by Trojani in 1984 [32], which will be discussed later, although there appears to be a growing recognition that not all of these categories or histological subtypes behave in exactly the same way, and that the same therapeutic approach is not necessarily optimal for all patients.

*(b)* 
*canine:*


In dogs, the group of STSs to which the grading scheme is typically applied includes various histological subtypes including fibrosarcoma (including the keloidal variant), myxosarcoma, liposarcoma, perivascular wall tumours, nerve sheath tumours (NST, non- plexus; also termed schwannomas, neurofibromas), pleomorphic sarcoma, undifferentiated sarcoma and malignant mesenchymoma [3]. There are others which are generally excluded from this group in terms of histological grading, including histiocytic sarcoma, lymphangiosarcoma, haemangiosarcoma, leiomyosarcoma, rhabdomyosarcoma and nerve sheath tumours arising from plexi, together with oral fibrosarcoma, gastrointestinal stromal tumours and synovial sarcoma. These are conventionally excluded from the group of STSs to which the grading system is applied because either they can be consistently distinguished by microscopic features and/or their anatomical location, and/or because they tend to exhibit more malignant biological behaviour when compared to those subtypes which are included [2,3,5,6,7]. More recently synovial sarcomas have been re-classified as peri-articular myxosarcoma or peri-articular histiocytic sarcoma [6]. This approach in canines is different to the one taken in humans; for dogs with STSs, this grouping becomes more a diagnosis of exclusion. 

In terms of relative prevalence, NSTs might be the most commonly occurring subtype [5], but over time the terminology used has been inconsistent and subject to change. For example, haemangiopericytoma was previously a frequently used diagnostic term but such tumours are now generally considered to be either perineural in origin or perivascular wall tumours [12]. Haemangiopericytomas most likely represent a small subset of the perivascular wall tumours, but without immunohistochemical staining these are difficult to differentiate from nerve sheath tumours [33].

Thus, it can be seen that the histological subtypes included in the STS group for human and canine patients varies, although there is overlap; there is also a difference in the prevalence of the different subtypes between humans and dogs. Thus the subgroup of STSs to which the grading is applied is not synonymous between the two species. 

*(c)* 
*feline:*


For cats, there is less data about specific histological subtypes and their relative frequencies, but the STS group would be likely to include NST, fibrosarcoma, myxosarcoma, leiomyosarcoma, liposarcoma, rhabdomyosarcoma, perivascular wall tumours and unspecified spindle cell tumours/sarcoma arising in the dermis or subcutis, regardless of their degree of differentiation [6]. This poses a few questions; cats presumably can develop perivascular wall tumours, but there is a lack of published data about them as a specific subtype. Tumours previously referred to as “giant cell tumours of soft tissues” have now been renamed as UPS, and appear to be included within the STS group [6]. FISS are incorporated in the feline STS group for the purposes of discussing histological grading.

## 6. Histological Grading of STS

*(a)* 
*Human:*


The grading system for STS used in humans originated from a study which looked at 155 patients with STSs of the various histological subtypes previously outlined [32]. In this study, seven different and individual criteria or parameters were analysed (tumour differentiation, tumour cellularity, atypical nuclei, malignant giant cells, mitotic count, tumour necrosis and vascular emboli) and assessed against outcomes (survival time, time to first metastasis and time to local recurrence). None of the seven criteria were significantly associated with time to local recurrence, and those significantly associated with the survival time and time to first metastasis were tumour differentiation, tumour cellularity, mitotic count, tumour necrosis and vascular emboli. Of these, the authors then individually scored tumour differentiation (scored 1 if well-differentiated, 2 if of uncertain histological type or 3 if embryonal, undifferentiated or of doubtful tumour type), mitotic count and tumour necrosis, with the total score correlating to a grade of either low (grade I), intermediate (grade II) or high (grade III), as shown in Table 1. However, it is important to note that the criterion which had the best prognostic value, specifically degree of tumour differentiation, is subjective in nature. Indeed the authors stated that “although the score system applied was made as simple as possible, the subjectivity of evaluation may introduce problems in the reproducibility of this grading” [32]. 

*(b)* 
*canine:*


This same grading system has been applied to dogs with STS in multiple studies [3,5,7]. One study found that canine STSs of higher histological grades were associated with a higher chance of local recurrence and within a reduced timeframe, however, the histological grade did not predict survival time [7]. Indeed, a comprehensive review of canine STS grading studies found that overall there does not appear to be a clear consensus on the association between histological grade and patient survival [3]. 

There are a number of suggested explanations for this. Firstly, many canines developing these tumours are older and may well die from other causes before metastatic disease or local recurrence becomes fatal [3]. Secondly, these tumours are relatively slow growing and often they can be managed locally without detrimental effects on quality of life. Even when metastatic lesions occur, they are also likely to be slow growing and thus may not impact on survival. 

This is quite different to humans where it was shown that 62.6% of STS patients developed metastatic disease, which accounted for 95.5% of the deaths; indeed, metastasis-free curves and survival time curves plotted according to tumour grade were strikingly similar to one another, highlighting the strong link between metastasis and survival time in human patients [32]. For dogs (and cats), local recurrence would appear to be a more frequent occurrence and the more likely cause of death when compared to humans with STS. Or at least, it is presumed to be; even when metastatic disease or local recurrence is suspected, it is very uncommon for it to be confirmed by biopsy and histopathological assessment in dogs (or cats). Of course, decisions made around “end of life” are rather different for veterinary patients. 

It is also important to note that the likelihood of local recurrence is dependent on upon the completeness of the surgical excision, but assessing the impact of this and disentangling it from any effect of the histological grade on outcome is fraught with difficulties; this is partly due to a lack of consistency when it comes to assessing surgical margins, and the terminology used, such as “close”, “narrow”, “clear”. As such, it may be that the impact of histological grade is most notable for patients with narrow or marginal excision of the tumour but further studies are needed [2,3,9,34,35,36]. 

*(c)* feline *injection* site sarcoma:

There have been several published studies which have applied the human/canine STS grading system in feline patients specifically diagnosed with FISS. One of these studies [37] examined three categories of tumours, including primary FISS (*n* = 44), FISS with local recurrence (*n* = 16) and non-vaccine associated fibrosarcomas (*n* = 10), however outcome data was only available for the locally recurrent FISS group and therefore the prognostic significance of the grade was not apparent [37]. Nevertheless, although there was a range of histological grades for each group of tumours, it is interesting to note there was a tendency for FISS to be of higher grade and the recurring FISS tumours had not assumed a more anaplastic phenotype relative to the primary FISS tumour group [37]. Another study in 2009 [38] was a retrospective analysis of radiation therapy as a treatment, in which a proportion of the tumours were graded (*n* = 55/73) and neither the histological grade nor the individual components were found to be predictive of survival or progression free interval, (nor was the Ki67 score, which also did not correlate with the histological grade) [38].

A more recent study by Porcellato et al. [39] divided 24 primary FISS cases into those with local recurrence (*n* = 10) or none (*n* = 14), with a minimum follow-up interval of 24 months. In this study the authors assessed a wide range of factors, including the histological grade, size of tumour, depth of infiltration, surgical margins, Ki67 score, and mitotic count. Two factors were found to be prognostic in terms of local recurrence, specifically the size of the tumour (after fixation, cut-off value of 3.75 cm) and the mitotic count (20 per 10 high power fields), which was also prognostic in relation to mortality, however the histological grade was not prognostic [39]. Thus on balance, there does not seem to be any evidence to support the usefulness of applying the human/canine grading system to FISS.

Some studies have focused on the immunohistochemical staining properties of FISS; one such study [40] included 21 cases, 18 classified as fibrosarcomas, and three as sarcomas. All tumours demonstrated positive staining for vimentin, 20 were positive staining for S100, and four for desmin. Thirteen of the 21 tumours demonstrated a degree of positive staining for Cox-2 and four for c-KIT. 

## 7. Other Feline Soft Tissue Sarcomas

Other than the studies described above which looked specifically at FISS, and the recent publication proposing a novel grading system for feline STSs [41], one other study looking at the prognostication of subtypes of feline STS is Schulman et al. [42]. In this study of 59 NSTs from 53 cats, there were three histologic categories. Sixteen tumours were categorised as histologically malignant; those 16 tumours had four or more mitotic figures in 10 high-power fields, and all bar one also had other histological features suggestive of malignancy, including hypercellularity, significant nuclear atypia, and tumour necrosis. The remaining tumours were classified as benign tumours with Antoni A areas that were S-100 and glial fibrillary acid protein (GFAP) positive (*n* = 34), or benign tumours that lacked Antoni A areas and were S-100 positive and GFAP negative (*n* = 9).

Alongside Schulman et al. [42], a few studies have looked at the immunohistochemical staining properties of this group of tumours in cats; such studies tend to focus on a particular histological subtype and/or single marker, trying to elucidate further the precise cell of origin for these tumours. One further study examined feline cutaneous NST in particular [43], with a total of 26 tumours of which 12 were classified as benign and 14 as malignant using similar criteria to Schulman et al. [42]. They reported all tumours as having diffuse expression of vimentin, whilst 25 (96.2%) expressed neuron-specific enolase (NSE), 24 (92.3%) expressed laminin and 25 (96.2%) expressed GFAP. Expression of S100 was more variable; overall 17 cases (65.4%) demonstrated some positive staining for S100, including both benign and malignant tumours (81.8% and 57.1% respectively), and with variable intensity. Five tumours also expressed smooth muscle actin (SMA), four of which were malignant. A case report published in 2018 [44] described a malignant cutaneous NST with rhabdomyosarcomatous differentiation (termed a triton tumour) in a cat. Immunohistochemical stains revealed diffuse strong expression of vimentin by both cell populations and multifocal cytoplasmic and nuclear staining for S100. For some markers, staining varied between areas, depending on cell differentiation; cells within some areas expressed desmin and less apparent S100 staining and no staining for GFAP or SMA. Desmin positive cells also expressed alpha sarcomeric actin and weak positive staining for myoglobin. 

A previous study [45] had also demonstrated immunoreactivity for KIT in 12 out of 46 cases (26%) of feline soft tissue fibrosarcomas, with four of those cases demonstrating positive staining in greater than 80% of tumour cells. This KIT staining was described as cytoplasmic and stippled within neoplastic spindle-shaped cells and/or multinucleate giant cells, but did not appear to correlate with the tumour being a FISS. 

Another recent study [46] examined feline giant cell pleomorphic sarcomas specifically (also known as undifferentiated pleomorphic sarcoma, previously known as malignant fibrous histiocytoma, anaplastic sarcoma with giant cells) with regards various features including their immunohistochemical staining properties. Neoplastic cells of both types, spindle cells and multinucleate giant cells, demonstrated positive staining for vimentin, as well as for ionized calcium-binding adaptor molecule 1 (Iba-1); spindle cells demonstrated strong positive cytoplasmic staining while the multinucleate giant cells demonstrated membranous and/or cytoplasmic staining. Staining patterns were variable for desmin and SMA, and staining for S100 was negative. 

## 8. Impact of Histological Grading on Prognostication of Feline STS

Given how common STSs are in cats, and that they are malignant with a range of potential biological behaviours, a grading system would be a valuable prognostic tool. To this end, a recent study aimed to produce a grading system specific for feline tumours. This retrospective study utilised a cohort of cats diagnosed with STS with a known clinical outcome [41]. Histological assessment was performed blinded to outcome and a set of potential criteria were each assessed individually, to identify those significantly associated with outcome in their own right. The features assessed for each tumour included mitotic count, presence of ulceration, necrosis (presence and extent), the size of the tumour, presence of any adjuvant-like material and multinucleate giant cells. Several of these criteria were then combined to produce a feline-specific grading system (Table 2 and Figure 1, Figure 2 and Figure 3). 

When applied to the tumours in the study, there were statistically significant differences in median survival time between cats with tumours of differing grades (Figure 4). When applied to the 47 cases with clinical outcome data available, 16 cats had grade I tumours. Of those 16, three were reported as having died as a result of tumour-related disease (TLR); in two cases treatment was not attempted and one had local recurrence with a survival time of 802 days. The median survival time (MST) was 900.5 days for cats diagnosed with grade I tumours. Fifteen of the 47 cats had grade II tumours. Of those, 8 were reported as having died as a result of TLR; six had local recurrence, one had suspected metastatic disease (not confirmed) and one was reported as tumour-related but the cause of death was not otherwise specified. The MST was 514 days for cats diagnosed with grade II tumours. Sixteen of the 47 cats had grade III tumours. Of those, 11 were reported as having died as a result of TLR; four had local recurrence, in four cases treatment was not attempted, three had suspected metastatic disease (not confirmed). The MST was 283 days for cats diagnosed with grade III tumours [41]. 

Importantly, although the inflammation score is still subjective, which is not ideal, it replaced another subjective component in the human/canine system (specifically, degree of differentiation) and a statistically significant association was found between the degree of inflammation and survival time [41]. 

## 9. Conclusions—Where Next for Feline Soft Tissue Sarcomas?

The grading scheme described above [41] is only a proposed system and as such there is now a need for a larger scale, preferably prospective studies to validate it fully, ideally via multicentre collaborations. There are drawbacks to the proposed grading scheme, including precisely which histological subtypes should be included within the STS group and whether FISS should be included or not. If we decide FISS should be addressed as a separate entity, we need to reach a consensus on just how we diagnose FISS with certainty. Reaching a consensus on such vital questions requires collaboration and standardisation, via such movements as the Veterinary Cancer Guidelines and Protocols group [47]. 

Another issue is the subjective nature of the inflammation score—although the differentiation score it replaces in the Trojani scheme [32] is also subjective, it would be advantageous if all criteria within any grading system were objective, readily obtainable from routinely-stained haematoxylin and eosin sections and easy to reproduce, thereby reducing variability between pathologists and laboratories [48,49]. With the advent and increasing adoption of image analysis within veterinary pathology [50] it may be that artificial intelligence plays an important role in quantifying criteria such as inflammation, as well as other features such as the extent of necrosis [51], and mitotic counts [52] with increased accuracy. 

Given the frequency of STSs in felines and the potential for malignant behaviour, more research is critical to give us a better understanding of this group of tumours. Large-scale, prospective studies are also needed to further validate the recently proposed grading system [41] and confirm its prognostic capabilities. Finally, more detailed investigations into the underlying genetics of this group of tumours are warranted, both to help better understand the biology of the disease and to aid in identifying potential diagnostic and prognostic markers as well as possible therapeutic targets. 

## Figures and Tables

**Figure 1 animals-12-02736-f001:**
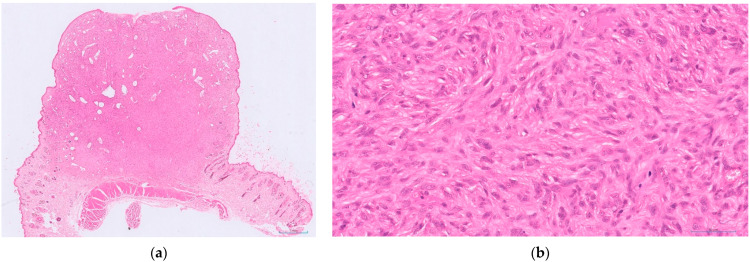
An example of a feline cutaneous soft tissue sarcoma of low histological grade (Grade I): (**a**) low power view of the tumour within haired skin, with focal ulceration (HE-stained; 1× magnification); (**b**): higher power view of the same tumour. This tumour had very mild inflammation (score 1), a mitotic count of 4 in 10 HPFs (400×; 2.37 mm^2^; score 1) and no necrosis (score 0)—giving a total score of 2 (HE-stained; 400× magnification).

**Figure 2 animals-12-02736-f002:**
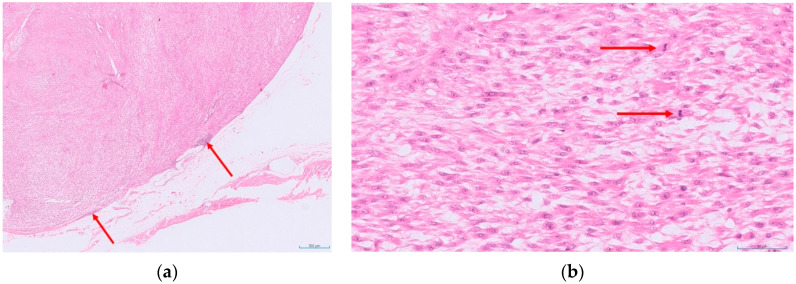
An example of a feline cutaneous soft tissue sarcoma of intermediate histological grade (grade II): (**a**) low power view of the tumour, with red arrows indicating very mild inflammation (score 1) with occasional focal lymphoid aggregates at the periphery only (HE-stained; 2× magnification); (**b**) a higher power view of the same tumour showing mitotic activity (red arrows indicate mitotic figures), with a mitotic count of 26 in 10 HPFs (400×; 2.37 mm^2^; score 3). This tumour contained some areas of necrosis but these comprised less than 50% of the total tumour present in the sections (score 1)—giving a total score of 5 (HE-stained; 400× magnification).

**Figure 3 animals-12-02736-f003:**
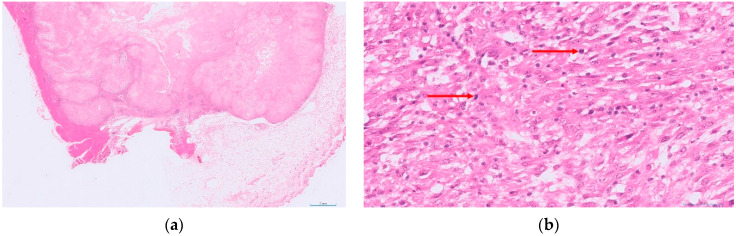
An example of a feline cutaneous soft tissue sarcoma of high histological grade (grade III): (**a**) low power view of the tumour with extensive central areas of necrosis and inflammatory cell infiltrates clearly visible (HE-stained; 5× magnification); (**b**) a higher power view of the same tumour, with severe inflammation (score 3), a mitotic count of 43 in 10 HPFs (400×; 2.37 mm^2^; score 3; red arrows highlight mitotic figures) and more than 50% necrosis (score 2)—total score 8 (HE-stained; 400× magnification).

**Figure 4 animals-12-02736-f004:**
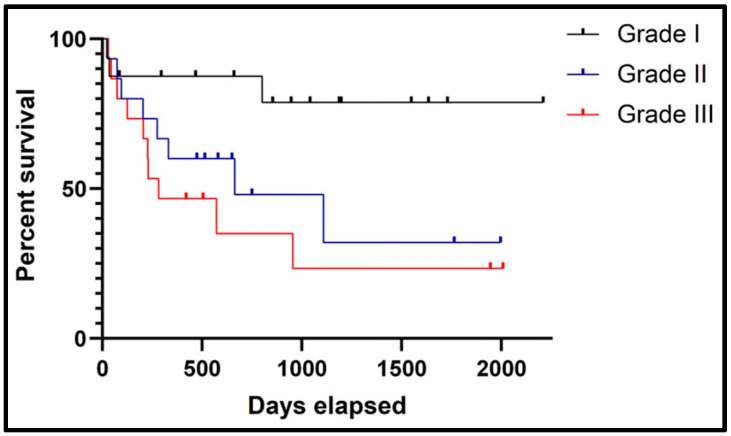
Kaplan–Meier survival curve for histological grade [41]. The black line represents cats with grade I (low grade) soft tissue sarcomas, the blue line represents cats with grade II (intermediate grade) soft tissue sarcomas and the red line represents cats with grade III (high grade) soft tissue sarcomas. Tick lines represent censored cases (lost to follow-up or died of non tumour-related causes).

**Table 1 animals-12-02736-t001:** Human soft tissue sarcoma grading system [32] as also routinely replied to canine soft tissue sarcomas.

**Differentiation Score**	
1	Sarcomas mostly resembling normal adult mesenchymal tissue
2	Sarcomas with known histological type but poor differentiation
3	Undifferentiated sarcomas, sarcomas of unknown type
**Mitotic Score ^1^**	
1	0–9
2	10–19
3	More than 19
**Tumour Necrosis Score**	
0	No necrosis
1	Equal to or less than 50% necrosis
2	More than 50% necrosis
**Histological Grade ^2^**	
I	Equal to or less than 3
II	4–5
III	Equal to or more than 6

^1^ Mitotic figures seen in 10 high power fields (400×). ^2^ total score when combining scores for differentiation, mitotic count and tumour necrosis.

**Table 2 animals-12-02736-t002:** Proposed grading system from feline cutaneous and subcutaneous soft tissue sarcomas [41].

**Inflammation Score**	
1	None, minimal or very mild
2	Mild to moderate
3	Severe
**Mitotic Score ^1^**	
1	0–9
2	10–19
3	More than 19
**Tumour Necrosis Score ^2^**	
0	No necrosis
1	Equal to or less than 50% necrosis
2	More than 50% necrosis
**Histological Gradem ^3^**	
I	Equal to or less than 3
II	4–5
III	Equal to or more than 6

^1^ Mitotic figures seen in 10 high power fields (400×). ^2^ Necrosis as a percentage of tumour area present in sections. ^3^ total score when combining scores for differentiation, mitotic count and tumour necrosis.

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
