# Peer review of "Feline Soft Tissue Sarcomas: A Review of the Classification and Histological Grading, with Comparison to Human and Canine"

_animals, 2022, doi:10.3390/ani12202736_

Round 1

Reviewer 1 Report

Overall, the manuscript is well written and provides an insight on an interesting and debated topic - sarcomas in cats. I personally find very interesting the comparison with humans and dogs.

However some concerns need to be address:

The introduction section sounds somehow vague, it does not provide sufficient references and arguments to introduce the reader to the topic. Furthermore, a clear aim has to be stated in this section. Although this is a review and not an original study, stating what are you reviewing and which will be your methods is anyhow useful for the reader. In the current form, it is not clear from the beginning whether the topic is "soft tissue sarcomas in general" or only STS without FISS. Given the existence of this two separate entities in cats, with clinically relevant differences in terms of therapy, prognosis and so on, I would highly recommend that in the introduction and in the following chapters it is stated which type of tutors will be discussed.

Another major concern is the paucity of bibliographic references. I would expect from a review that all the most relevant references are cited - I feel this is not the case. I would suggest improving the bibliography. This would also help to improve the manuscript, as some information are lacking or presented in a vague manner.

L156: I would suggest avoiding putting the name of an author in the title of a subheading: please delate "Trojani

L208: add reference

L212-285: this is all about cats. I would just leave the heading "feline" and add subheadings: at L213 "Feline Injection Site Sarcoma", at L236 "Other feline sarcomas"

L266-274: move to FISS section (the one suggested in the previous comment)

L286: please consider amending this subheading to: Relevance of grading system for prognostication of feline sarcomas; or something similar of your choice.

L336: not instead of now

L344-347: please consider delating this part or if you want to keep it you should go into more details and add some references.

Author Response

Responses to reviewer’s comments

Overall, the manuscript is well written and provides an insight on an interesting and debated topic - sarcomas in cats. I personally find very interesting the comparison with humans and dogs. However some concerns need to be address

The introduction section sounds somehow vague, it does not provide sufficient references and arguments to introduce the reader to the topic. Furthermore, a clear aim has to be stated in this section. Although this is a review and not an original study, stating what are you reviewing and which will be your methods is anyhow useful for the reader. In the current form, it is not clear from the beginning whether the topic is "soft tissue sarcomas in general" or only STS without FISS. Given the existence of this two separate entities in cats, with clinically relevant differences in terms of therapy, prognosis and so on, I would highly recommend that in the introduction and in the following chapters it is stated which type of tutors will be discussed.

  • We have added a new introductory paragraph which outlines the contents of the review, and which types of tumour will be discussed. The section on terminology is now separate. We hope this makes things clearer for the reader.

Another major concern is the paucity of bibliographic references. I would expect from a review that all the most relevant references are cited - I feel this is not the case. I would suggest improving the bibliography. This would also help to improve the manuscript, as some information are lacking or presented in a vague manner.

  • Additional references have been added to various parts of the manuscript, however there is a general lack of published studies looking at soft tissue sarcomas in cats.

L156: I would suggest avoiding putting the name of an author in the title of a subheading: please delate "Trojani

  • This has been deleted

L208: add reference

  • These have been added

L212-285: this is all about cats. I would just leave the heading "feline" and add subheadings: at L213 "Feline Injection Site Sarcoma", at L236 "Other feline sarcomas"

  • This has been done

L266-274: move to FISS section (the one suggested in the previous comment)

  • This has been done

L286: please consider amending this subheading to: Relevance of grading system for prognostication of feline sarcomas; or something similar of your choice.

  • This has been amended

L336: not instead of now

  • Thank you for highlighting this error, it has now been corrected

L344-347: please consider delating this part or if you want to keep it you should go into more details and add some references.

  • We have added in additional references but going into more details is not within the main scope of this review – hopefully the readers will find the references of interest. Since this is really part of the conclusions, it is also somewhat speculative regarding the future directions this area might take, rather than what has already been published

Reviewer 2 Report

In this review, the author explains which histological subtypes of tumors are gathered under the designation “soft tissue sarcomas” in cats, dogs and humans, which grading scheme is applied to canine and human STSs, and which one is proposed for feline STSs, and also focus on the unresolved questions regarding feline injection-site sarcomas, the role of immunohistochemical markers, and the subjectivity of histological grading. This review is very well written and very informative, the manuscript is clear and concise, and well illustrated. 

In my opinion, this manuscript is suitable for publication in Animals, although I have suggested minor recommendations, especially regarding some paragraphs that do not mention the corresponding literature.

Minor comments

-       Page 1, lines 40-44, introduction: please add an appropriate reference(s) for the use of “soft issue tumours” instead of “soft tissue sarcomas”, and/or for the negative connotation of “STS” for veterinary practitioners. A reference can also be added page 2, line 48, to support the idea that low-grade soft tissue sarcomas tend not to metastasize.

-       Page 2, lines 60-63, introduction: please add appropriate reference(s) on the immunohistochemical markers that can be used to subtype soft tissue sarcomas, at least in dogs and cats. 

-       Page 2, line 65: in this chapter “2 STSs in cats, prevalence and clinical behavior”, lines 66-84 actually correspond to STS prevalence in cats, while lines 85-98 deal with clinical behavior in dogs and cats, so maybe this chapter could be divided into 2 parts. If the purpose of this review is to compare STSs in cats, dogs, and humans, maybe the clinical behavior of human STSs could be summarized here.

-       Page 2, lines 78-82, STSs in cats: if the author choose to make mention of feline STSs under the age of 12 months, maybe a mention on the role of FeSV could be added.

-       Page 4, lines 147-153, histological subtypes of feline STSs: please add the corresponding references. Lines 153-155, former “giant cell tumours of soft tissues”, please reformulate the sentence as an affirmation, not a question. 

-       Page 5, lines 204-208, risk of local recurrence according to the margin status and definitions of the margin status: please add the corresponding references.

-       Page 7, lines 297-298, feline STS prognosis: here, it would be nice to summarize the survival probabilities associated with the histological grades. 

-       Figures 1, 2, and 3: please add scale bars. In Figure 1, panel (b) may not be necessary. Figure 2, panel (b): two mitoses could be designated using arrows. Figure 3, panel (b): is it possible to replace it by a picture taken at 400x magnification? 

Typos / Spelling

-       Page 1, line 21, abstract: “by” instead of “but”.

-       Page 1, lines 29-31, introduction: the use of “we” in a review can be considered familiar language, and could be avoided. Same page 3, line 100. Same page 4, line 147. Same page 9, lines 332-337.

-       Page 2, line 74, STSs in cats: “[“ instead of “(“ before references 4,6,7.

-       Page 2, line 83, STSs in cats: maybe “British Shorthair” is more appropriate than “British Blue” for this breed.

-       Page 4, line 179, footnote 2 of Table 1: please add “score” or “count” after “mitotic”. Same page 5, line 307.

-       Page 9, line 336, conclusions: “not” instead of “now”.

Author Response

Response to reviewer’s comments

In this review, the author explains which histological subtypes of tumors are gathered under the designation “soft tissue sarcomas” in cats, dogs and humans, which grading scheme is applied to canine and human STSs, and which one is proposed for feline STSs, and also focus on the unresolved questions regarding feline injection-site sarcomas, the role of immunohistochemical markers, and the subjectivity of histological grading. This review is very well written and very informative, the manuscript is clear and concise, and well-illustrated. 

In my opinion, this manuscript is suitable for publication in Animals, although I have suggested minor recommendations, especially regarding some paragraphs that do not mention the corresponding literature.

  • Thank you for your positive comments about the manuscript and also for the recommendations; we hope that we have addressed them all as per suggestions, and feel that they have indeed strengthened the review.

Minor comments

-       Page 1, lines 40-44, introduction: please add an appropriate reference(s) for the use of “soft issue tumours” instead of “soft tissue sarcomas”, and/or for the negative connotation of “STS” for veterinary practitioners. A reference can also be added page 2, line 48, to support the idea that low-grade soft tissue sarcomas tend not to metastasize.

  • We have included further references for the use of soft tissue tumours as a term, and for low grade STS and metastatic rates. We have removed the sentence about negative connotations for veterinary practitioners

-       Page 2, lines 60-63, introduction: please add appropriate reference(s) on the immunohistochemical markers that can be used to subtype soft tissue sarcomas, at least in dogs and cats. 

  • We have included a selection of references which use IHC in part of their investigations into canine STS, although IHC for feline STS is covered later in the review and so we have left those where they already are. We hope this is OK.

-       Page 2, line 65: in this chapter “2 STSs in cats, prevalence and clinical behavior”, lines 66-84 actually correspond to STS prevalence in cats, while lines 85-98 deal with clinical behavior in dogs and cats, so maybe this chapter could be divided into 2 parts. If the purpose of this review is to compare STSs in cats, dogs, and humans, maybe the clinical behavior of human STSs could be summarized here.

  • Have divided into two chapters and renamed accordingly – thank you for highlighting this.

-       Page 2, lines 78-82, STSs in cats: if the author choose to make mention of feline STSs under the age of 12 months, maybe a mention on the role of FeSV could be added.

  • This has been added, thank you for raising this

-       Page 4, lines 147-153, histological subtypes of feline STSs: please add the corresponding references.

  • We have added in the reference of the CL Davis Fascicle here, although there is a general absence of discussion around this group of tumours specifically for cats

Lines 153-155, former “giant cell tumours of soft tissues”, please reformulate the sentence as an affirmation, not a question. 

  • We have rewritten this so it is no longer a question

-       Page 5, lines 204-208, risk of local recurrence according to the margin status and definitions of the margin status: please add the corresponding references.

  • Additional references have been added

-       Page 7, lines 297-298, feline STS prognosis: here, it would be nice to summarize the survival probabilities associated with the histological grades. 

  • We have added additional information with regards survival times and K-M curve

-       Figures 1, 2, and 3: please add scale bars. These have been added.

In Figure 1, panel (b) may not be necessary – this has been removed

Figure 2, panel (b): two mitoses could be designated using arrows. This has been done.

Figure 3, panel (b): is it possible to replace it by a picture taken at 400x magnification? This has been done.

Typos / Spelling

-       Page 1, line 21, abstract: “by” instead of “but”.

  • Thank you for pointing out this typographical error. It has now been corrected.

-       Page 1, lines 29-31, introduction: the use of “we” in a review can be considered familiar language, and could be avoided. Same page 3, line 100. Same page 4, line 147. Same page 9, lines 332-337.

  • These sentences have now been rewritten

-       Page 2, line 74, STSs in cats: “[“ instead of “(“ before references 4,6,7.

  • Thank you for pointing out this typographical error, much appreciated. It has now been corrected

-       Page 2, line 83, STSs in cats: maybe “British Shorthair” is more appropriate than “British Blue” for this breed.

  • Whilst we agree that British Shorthair would be more appropriate than British Blue, this is the breed originally reported in the study referenced here, so it does not seem correct to change this for the review.

-       Page 4, line 179, footnote 2 of Table 1: please add “score” or “count” after “mitotic”. Same page 5, line 307.

  • This has now been amended, thank you

-       Page 9, line 336, conclusions: “not” instead of “now”.

  • Thank you for pointing out this typographical error. It has now been corrected

Round 2

Reviewer 1 Report

Thank you for addressing my concerns. Just some minor comments.

Title: please change ; to :

L30, L31, L45, L49, : can you please reference this two statements?

L53: add Kuntz 1997 as reference here

L61: I would suggest adding some details of what is it the DVM foundation, as it may not be know by all readers.

L110: please add mention of the fact that several factors have been proposed to predict the risk of local recurrence in FISS and STS  in general (both in dogs and cats), including resection margins, histological grading, size and location of tumor, expertise of surgeon, recurrent tumours, p53 expression, mitotic count and differentiation, neutrophil-to-leukocyte ratio. Please also mention that, despite those factors, there is still a subset of tumors for which prediction of local recurrence is inaccurate. Please also add appropriate references for all the above-mentioned prognostic factors for local recurrence.

L311: I would suggest changing the subheading to: "impact of grading on prognostication of feline STS", otherwise it seems to overlap with paragraph 4.

Author Response

Title: please change ; to :

  • This has now been done

L30, L31, L45, L49, : can you please reference this two statements?

  • This has now been done

L53: add Kuntz 1997 as reference here

  • This has now been done

L61: I would suggest adding some details of what the DVM foundation is, as it may not be known by all readers.

  • This has now been done, and I have added a link to the website

L110: please add mention of the fact that several factors have been proposed to predict the risk of local recurrence in FISS and STS in general (both in dogs and cats), including resection margins, histological grading, size and location of tumor, expertise of surgeon, recurrent tumours, p53 expression, mitotic count and differentiation, neutrophil-to-leukocyte ratio. Please also mention that, despite those factors, there is still a subset of tumors for which prediction of local recurrence is inaccurate. Please also add appropriate references for all the above-mentioned prognostic factors for local recurrence.

  • Local recurrence is discussed later in the review, on lines 225-231; this section was added following the previous round of reviews and suggestions from reviewers.
  • Whilst we agree that local recurrence and margin assessment are very important topics, the focus of this particular review is the histological grading of these tumours.
  • In the new additional paragraph, we have expanded the references provided so hopefully readers can follow those up if they want more detail specifically regarding margins and local recurrence risks.

L311: I would suggest changing the subheading to: "impact of grading on prognostication of feline STS", otherwise it seems to overlap with paragraph 4.

  • This has now been done